# Spacecraft Segment Damage Identification Method Based on Fiber Optic Strain Difference Field Reconstruction and Norm Calculation

**DOI:** 10.3390/s23218822

**Published:** 2023-10-30

**Authors:** Jihong Xu, Jie Zeng, Binbin Chen, Ruixin Lu, Yangyang Zhu, Lei Qi, Xiangfei Chen

**Affiliations:** 1College of Aerospace Engineering, Nanjing University of Aeronautics and Astronautics, Nanjing 210016, China; xujihong@nuaa.edu.cn (J.X.); cbb2000@nuaa.edu.cn (B.C.); lurx127@nuaa.edu.cn (R.L.); sz2201125@nuaa.edu.cn (Y.Z.); 2Beijing Institute of Spacecraft Environment Engineering, Beijing 100094, China; qilei511@126.com; 3College of Engineering and Applied Sciences, Nanjing University, Nanjing 210023, China; chenxf@nju.edu.cn

**Keywords:** spacecraft segment structure, fiber optic grating sensor, strain response difference field, vector norms, damage recognition

## Abstract

Real-time online identification of spacecraft segment damage is of great significance for realizing spacecraft structural health monitoring and life prediction. In this paper, a damage response characteristic field inversion algorithm based on the differential reconstruction of strain response is proposed to solve the problem of not being able to recognize the small damages of spacecraft structure directly by the strain response alone. Four crack damage location identification methods based on vector norm computation are proposed, which realize online identification and precise location of structural damage events without external excitation by means of spacecraft structural working loads only. A spacecraft segment structural damage monitoring system based on fiber optic grating sensors was constructed, and the average error of damage localization based on the curvature vector 2 norm calculation was 2.58 mm, and the root-mean-square error was 1.98 mm. The results show that the method has superior engineering applicability for on-orbit service environments.

## 1. Introduction

On-orbit spacecraft are inevitably subject to complex impact loads such as micrometeoroid and space debris impacts, which lead to cracks through holes and other small damages in the spacecraft segment structure and leakage, thus affecting the normal conduct of space missions [1,2]. Therefore, it is of great significance to realize real-time online monitoring of spacecraft structural damage [3].

Conventional non-destructive testing methods used for damage identification include the infrared imaging method, acoustic emission detection method and so on. The infrared imaging method is realized by detecting the size of the heat radiated from the surface of the object under test [4], which is able to realize the damage detection over a large area, but when the structure undergoes violent vibration in service, it will lead to shaking of the infrared camera, which will result in errors in the detection results. Acoustic emission detection method is based on the transient elastic wave generated by the rapid release of local energy for damage location identification [5]; this method has the advantages of higher sensitivity and shorter detection time, but it has obvious timeliness, and once the response information of the structure at the time of the damage is missed or lost, the structure’s health status cannot be effectively and objectively evaluated.

The structural health-monitoring method based on piezoelectric sensing array excites and receives guided wave signals online through piezoelectric devices integrated inside or on the surface of the structure, and identifies the structural damage state based on the change in the guided wave signal characteristics [6,7], but it requires a large number of signal transmission cables, which is inclined to cause complexity of the monitoring system, and there are certain limitations in monitoring structural damage at large scales. In contrast, distributed fiber optic sensors have unique advantages such as resistance to electromagnetic and radio frequency interference, integration of transmission and sensing, and good applicability for real time monitoring of spacecraft structural damage [8,9]. Fiber optic sensors include Fiber Bragg Grating sensors (FBGs) and Optical Frequency Domain Reflectometry (OFDR). OFDR has high resolution and measurement accuracy, and is suitable for short-range, high-precision distributed strain and temperature measurement [10,11,12]. Compared with OFDR sensor, FBGs’ sampling frequency is higher, which is conducive to better acquisition of dynamic damage response information caused by dynamic shock or vibration, and the construction cost of the FBG demodulation system is lower; in addition, FBGs demodulation module is convenient to build a quasi-distributed monitoring system with dozens of synchronous sampling channels. Therefore, this paper selects FBGs for damage identification of compartment structure.

In addition to various monitoring techniques, damage identification and localization algorithms are also key technologies to achieve spacecraft structural health status assessment. Currently, researchers have conducted many studies on structural damage monitoring methods based on fiber optic sensors [13,14].

Xu et al. [15] constructed an impact response sample database of damage containing quadrilateral solid supported aluminum-alloy thin-plate structure through external knocking experiments, and proposed a structural damage location identification algorithm based on the combination of wavelet packet energy spectra and BP neural network. Jia et al. [16] realized the leakage localization of the pipeline by collecting pipeline circumferential strains as the input sample dataset for the support vector regression algorithm. Ni et al. [17] proposed a Bayesian machine learning method based on the training of rail strain response data during train passage to realize the online monitoring of wheel defects.

Relevant scholars have also carried out a series of research on structural damage localization methods based on external excitation and dynamic norm identification. Dimitrios Anastasopoulos et al. [18] realized the identification of structural damage location of bridge structure by adopting external impact load as excitation, and obtaining the information of the change in strain mode norms before and after the bridge damage. Based on the external excitation, Wang et al. [19] studied a method of pipeline structural health monitoring, calculated the weighted intrinsic frequency change rate and strain modal change rate caused by structural crack damage, and realized the structural crack damage localization.

In addition, a high-density placement of sensors method can also be used to identify the damage location of severely damaged structures. For example, Dimitrios P Milanoskiv et al. [20] proposed a damage localization discriminant index for debonding of skinned wall panels and bars by placing 10 fiber grating sensors on the same bar with high spatial resolution, and localized the damage area of debonded bars according to the change in strain response of the structure where the sensors were located caused by the damage. Sidney Goossens et al. [21] proposed a damage discrimination index based on continuous measurement of high spatial resolution strain, and realized the damage localization of impact-induced debonding of composite reinforced wall panel bars by high-density surface applied 60 fiber grating sensors on two bars with a length of 405 mm.

Nevertheless, there is still room for improvement in the structural damage identification method based on fiber optic sensors in the field of practical aerospace engineering. The model identification method needs to construct a database containing a large number of damage samples for different spacecraft structures and damage forms, which is costly and time consuming. Once the structural form or loading conditions change, the previously constructed damage identification model will not be applicable. Another type of method requires external excitation to obtain the characteristic changes of norms such as intrinsic frequency, displacement mode, and strain mode of the object under test due to damage, whereas it is usually difficult to apply external excitation during spacecraft operation in orbit, so it is necessary to develop crack damage localization algorithms for spacecraft structures without external active excitation. In addition, the actual service conditions require more online identification and accurate localization of Barely Visible Impact Damage (BVID) [22] small crack damage, and the strain distribution information obtained only through the high-density continuous arrangement of strain sensors not only greatly increases the burden and complexity of the monitoring system, but also often fails to identify the small changes in structural strains due to localized small crack damages with a high degree of sensitivity.

As for the problems mentioned above, this paper proposes a damage response feature field inversion algorithm based on discrete strain information sensing, which adopts the difference of strain response as the input feature for damage feature field reconstruction, and solves the difficult problem of failing to recognize the small crack damages of spacecraft structure directly based on the distribution of the strain response alone. The method can obtain the crack damage feature distribution equivalent to the global actual measurement without the need of high-density arrangement of sensors, which realizes the identification of structural crack damage events and regional localization without external excitation, and improves the engineering applicability of the algorithm for on-orbit service conditions. On this basis, four methods are proposed to accurately locate the structural crack damage coordinates of spacecraft segments based on vector norm calculations, which break through the limitations of time-consuming and labor-intensive, structural damage susceptibility, and limited applicability caused by the need to construct a large scale damage sample library to improve the localization accuracy of the pattern recognition methods.

## 2. Materials and Methods

### 2.1. Structural Damage Equivalency Method for Compartmentalized Structures

A typical spacecraft segment structure is equated to a linear system with multiple degrees of freedom. The dynamic equations of this structure before and after damage occurs are expressed as [23]
(1)$$\left[M_{s0}\right]\left\{{\ddot{x}}_{s0}\right\}+\left[C_{s0}\right]\left\{{\dot{x}}_{s0}\right\}+\left[K_{s0}\right]\left\{x_{s0}\right\}=\left\{f_{s}\right\}$$
(2)$$\left[M_{s0}\right]\left\{{\ddot{x}}_{s1}\right\}+\left[C_{s0}\right]\left\{{\dot{x}}_{s1}\right\}+\left[K_{s1}\right]\left\{x_{s1}\right\}=\left\{f_{s}\right\}$$

On the one hand, the mass and damping will not change significantly after the structural damage occurs, and the small changes can be ignored. On the other hand, load is an external factor that does not vary with structural damage. In addition, compared with the complete structure, the stiffness of the structure with damage will change significantly, because cracks, through holes and other damage, will destroy the integrity of the structure, resulting in the reduction of the bearing area, and then lead to the weakening of the bearing capacity.

Where [*M_s_*_0_] is the mass matrix of the segment structure. [*C_s_*_0_] is the damping matrix of the segment structure. [*K_s_*_0_] is the structural stiffness matrix of the undamaged segment structure. [*K_s_*_1_] is the structural stiffness matrix of the damage containing segment structure. {*x_s_*_0_}, {*ẋ*_s0_}, and {$\ddot{x}$*_s_*_0_} are the structural displacements, velocities, and accelerations of the undamaged compartment, respectively. {*x_s_*_1_}, {*ẋ*_s1_}, and {$\ddot{x}$*_s_*_1_} are the structural displacement, velocity, and acceleration of the undamaged compartment, respectively. {*f_s_*} is the vector of the applied external loads before and after the occurrence of damage.

According to the Laplace variation, let {*f_s_*} = {*F_s_*}*e^jω^*, {*x_s_*_0_} = {*U_s_*_0_}*e^jω^*, and {*x_s_*_1_} = {*U_s_*_1_}*e^jω^*. Then, the kinetic equations before and after the occurrence of damage to the spacecraft segment structure are transformed to [24]
(3)$$\left(-\omega^{2}\left[M_{s0}\right]+j\omega \left[C_{s0}\right]+\left[K_{s0}\right]\right)\left\{U_{s0}\right\}=\left\{F_{s}\right\}$$
(4)$$\left(-\omega^{2}\left[M_{s0}\right]+j\omega \left[C_{s0}\right]+\left[K_{s1}\right]\right)\left\{U_{s1}\right\}=\left\{F_{s}\right\}$$
where *e* is a natural constant. *ω* is the intrinsic frequency of the compartment structure. *j* is an imaginary unit. {*U_s_*_1_} is the displacement vector of the damage-containing structure. {*U_s_*_0_} is the displacement vector of the damage-free structure, and {*F_s_*} is the structural load vector.

The following relationship exists between the displacement vectors of damage-free and damage-containing structures:(5)$$\left\{U_{s1}\right\}=\left\{U_{s0}\right\}+\left\{\Delta U_{s}\right\}$$
where {Δ*U_s_*} is the change in structural displacement.

The relationship exists between structural unit displacement and strain [25]:(6)$$\left\{\varepsilon \right\}=\left[B\right]\left\{U_{s}\right\}$$
where {*ε*} is the structural strain matrix. [*B*] is the structural cell geometry matrix, and {*U_s_*} is the structural displacement matrix.

Then, the following functional relationship exists between the strain of the damage-free structure and the strain of the damage-containing structure:(7)$$\left\{\varepsilon_{s1}\right\}=\left\{\varepsilon_{s0}\right\}+\left\{\Delta \varepsilon_{s}\right\}$$
where {*ε_s_*_1_} is the structural strain with damage. {*ε_s_*_0_} is the structural strain without damage. {Δ*ε_s_*} is the change in structural strain.

The following relationship exists between the stiffness matrix of the undamaged structure and the structure containing crack damage:(8)$$\left\{K_{s1}\right\}=\left\{K_{s0}\right\}+\left\{\Delta K_{s}\right\}$$
where {Δ*K_s_*} is the structural stiffness change matrix.

Combining Equations (3)–(8), the simplification gives the relationship between the structural strain change and the concentrated additional load as:(9)$$\left(-\omega^{2}\left[M_{s0}\right]+j\omega \left[C_{s0}\right]+\left[K_{s0}\right]\right)\left\{\Delta \varepsilon \right\}=-\left[\Delta K_{s}\right]\left\{\varepsilon_{s1}\right\}=\left\{\Delta F\right\}$$
where {Δ*F*} is the centralized additional load, then from Equation (9), it can be known that the structural damage caused by the strain response difference characteristic field mutation effect can be equated to the damage region in the application of centralized additional load, which is defined in this paper as the “damage equivalent load”.

### 2.2. Damage Region Localization Method Based on Strain Response Difference Field Inversion

According to the structural dynamic response equations, for the strain response induced by multiple loads acting simultaneously on a compartmentalized structure, it can be equated to a linear superposition of the strain response induced by individual loads acting separately on that structure, with the relationship expressed as [26]:(10)$$\left[M\right]\left\{\ddot{x}\right\}+\left[C\right]\left\{\dot{x}\right\}+\left[K\right]\left\{x\right\}=\left\{F\right\}$$
(11)$$\left\{F\right\}=\left\{F_{1}\right\}+\left\{F_{2}\right\}+\left\{F_{3}\right\}+\cdots \cdots +\left\{F_{r}\right\}$$
(12)$$\left[M\right]\left\{{\ddot{x}}_{1}\right\}+\left[C\right]\left\{{\dot{x}}_{1}\right\}+\left[K\right]\left\{x_{1}\right\}=\left\{F_{1}\right\}$$
(13)$$\left[M\right]\left\{{\ddot{x}}_{2}\right\}+\left[C\right]\left\{{\dot{x}}_{2}\right\}+\left[K\right]\left\{x_{2}\right\}=\left\{F_{2}\right\}$$
(14)$$\left[M\right]\left\{{\ddot{x}}_{3}\right\}+\left[C\right]\left\{{\dot{x}}_{3}\right\}+\left[K\right]\left\{x_{3}\right\}=\left\{F_{3}\right\}$$
⋮
(15)$$\left[M\right]\left\{{\ddot{x}}_{r}\right\}+\left[C\right]\left\{{\dot{x}}_{r}\right\}+\left[K\right]\left\{x_{r}\right\}=\left\{F_{r}\right\}$$
where [*M*] is the structural quality matrix. [*C*] is the structural damping matrix. [*K*] is the structural stiffness matrix. {*F*_1_}, {*F*_2_}, {*F*_3_}, ……, {*F_r_*} indicates that {*F*} is decomposed into *r* individual action loads, and {*F_r_*} is the *r*th load. {*x*_1_}, {*x*_2_}, {*x*_3_}, ……, {*x*_r_} are displacement vectors, {*ẋ*_1_}, {*ẋ*_2_}, {*ẋ*_3_}, ……, {*ẋ*_r_} are velocity vectors, and {$\ddot{x}$_1_}, {$\ddot{x}$_2_}, {$\ddot{x}$_3_}, ……, {$\ddot{x}$_r_} are acceleration vectors, all under the action of r individual loads alone.

Therefore, the expression for the linear superposition displacement response of the compartment structure is
(16)$$\left\{x\right\}=\left\{x_{1}\right\}+\left\{x_{2}\right\}+\left\{x_{3}\right\}+\cdots +\left\{x_{r}\right\}$$

According to the Laplace transform, the expression for the linear superposition strain response of the compartment segment structure is solved as
(17)$$\left\{\varepsilon \right\}=\left\{\varepsilon_{1}\right\}+\left\{\varepsilon_{2}\right\}+\left\{\varepsilon_{3}\right\}+\cdots +\left\{\varepsilon_{r}\right\}$$
where {*ε*_1_}, {*ε*_2_}, {*ε*_3_}, ⋯⋯, {*ε_r_*} denotes the strain response vectors under *r* loads alone; {*ε_r_*} is the strain response vector under the *r*th load {*f_sr_*}.

For the same strain measurement point location of the segment structure before and after damage, the strain difference vector at the location of the point caused by the concentrated additional load (i.e., damage equivalent load) is
(18)$${\left\{\Delta \varepsilon \right\}}_{M}={\left\{\varepsilon_{1}\right\}}_{M}-{\left\{\varepsilon_{0}\right\}}_{M}$$
where {*ε*_1_}*_M_* is the strain vector at the point where the fiber grating sensor is affixed to the damage-containing structure. {*ε*_0_}*_M_* is the strain vector at the point where the fiber grating sensor is affixed to the damage-free structure, and {Δ*ε*}*_M_* is the vector of the difference in strains at the position where the fiber grating sensor is affixed due to the concentrated additional load used to simulate the equivalent damage.

When the nacelle segment structure is under working load, according to the principle of linear superposition of strains, the strain vector at the measurement point can be expressed as [27]
(19)$${\left\{\varepsilon \right\}}_{M}=\left[\begin{array}{llll} \delta_{1}{}^{1} & \delta_{2}{}^{1} & \cdots & \delta_{p}{}^{1}\\ \delta_{1}{}^{2} & \delta_{2}{}^{2} & \cdots & \delta_{p}{}^{2}\\ \vdots & \vdots & \ddots & \vdots \\ \delta_{1}{}^{M} & \delta_{2}{}^{M} & \cdots & \delta_{p}{}^{M} \end{array}\right]\;\cdot \;\left[\begin{array}{l} \omega_{1}\\ \omega_{2}\\ \;\vdots \\ \omega_{p} \end{array}\right]={\left[\delta_{{}_{0}}\right]}_{M\times p\;}\cdot \;{\left\{\omega \right\}}_{p}$$
where {*ε*}*_M_* is the strain vector of the measurement point where the fiber grating sensor is located, *p* is the number of applied damage equivalent loads, [δ_0_]*_M×p_* is the load–strain matrix corresponding to the measurement point under the action of *p* damage equivalent loads alone, {*ω*}*_p_* is the load weighting coefficient, and $\delta_{p}{}^{M}$ is expressed as the value of the strain response of the *M*th measurement point under the action of the *p*th damage equivalent load alone.

When the structure is under external loading, the strain vector at the inversion point can be expressed according to the principle of linear superposition of strains:(20)$${\left\{\varepsilon \right\}}_{N}=\left[\begin{array}{llll} \delta_{1}{}^{1} & \delta_{2}{}^{1} & \cdots & \delta_{p}{}^{1}\\ \delta_{1}{}^{2} & \delta_{2}{}^{2} & \cdots & \delta_{p}{}^{2}\\ \vdots & \vdots & \ddots & \vdots \\ \delta_{1}{}^{N} & \delta_{2}{}^{N} & \cdots & \delta_{p}{}^{N} \end{array}\right]\;\cdot \;\left[\begin{array}{l} \omega_{1}\\ \omega_{2}\\ \;\vdots \\ \omega_{p} \end{array}\right]={\left[\delta_{{}_{0}}\right]}_{N\times p\;}\cdot \;{\left\{\omega \right\}}_{p}$$
where {*ε*}*_N_* is the strain vector of the inversion point. [δ_0_]*_N×p_* is the load strain matrix corresponding to the inversion point under *p* damage equivalent loads acting individually, and $\delta_{p}{}^{N}$ is expressed as the value of the strain response of the *N*th inversion point under *p* damage equivalent loads acting individually.

If *M ≤ N*, then the expression is obtained by solving Equations (19) and (20):(21)$${\left\{\varepsilon \right\}}_{N}={\left[\delta_{{}_{0}}\right]}_{N\times p}{\left({\left[\delta_{{}_{0}}\right]}_{M\times p}{}^{T}{\left[\delta_{{}_{0}}\right]}_{M\times p}\right)}^{-1}{\left[\delta_{{}_{0}}\right]}_{M\times p}{}^{T}{\left\{\varepsilon \right\}}_{M}$$

The strain vector {*ε*_1_}*_M_* of the measured points on the surface of the structure of the damage-containing compartment segment is brought to Equation (21), and the inversion yields the strain response field as:(22)$${\left\{\varepsilon_{1}\right\}}_{N}={\left[\delta_{{}_{0}}\right]}_{N\times p}{\left({\left[\delta_{{}_{0}}\right]}_{M\times p}{}^{T}{\left[\delta_{{}_{0}}\right]}_{M\times p}\right)}^{-1}{\left[\delta_{{}_{0}}\right]}_{M\times p}{}^{T}{\left\{\varepsilon_{1}\right\}}_{M}$$
where {*ε*_1_}*_N_* is the surface strain response field of the damage-containing compartment structure.

The strain response characteristic difference field {Δ*ε*}*_M_* is obtained by inverting the vector of strain differences at the structural surface points into Equation (21):(23)$${\left\{\Delta \varepsilon \right\}}_{N}={\left[\delta_{{}_{0}}\right]}_{N\times p}{\left({\left[\delta_{{}_{0}}\right]}_{M\times p}{}^{T}{\left[\delta_{{}_{0}}\right]}_{M\times p}\right)}^{-1}{\left[\delta_{{}_{0}}\right]}_{M\times p}{}^{T}{\left\{\Delta \varepsilon \right\}}_{M}$$
where {Δ*ε*}*_N_* is the characteristic difference field of the surface strain response of the compartment structure obtained from the inversion.

According to the uncertain existence of a sudden change in the field of the characteristic difference of the strain response, it is possible to realize the identification of the structure with or without damage. According to the area of the sudden change in the characteristic difference of the strain response, it is possible to realize the identification of the number of structural damages as well as the damage area.

The flow of the strain response difference field inversion algorithm based on the load–strain linear superposition algorithm is shown in Figure 1.

### 2.3. Damage Coordinate Identification Method Based on Norm Calculation

By analyzing the mutation characteristics of the strain response difference field, the number of damages and the approximate area of damage can be determined, but the precise location coordinates of the damage cannot be given. In order to further achieve precise localization of damage in the structure, four indicators were introduced: strain vector 1 norm, strain vector 2 norm, curvature vector 1 norm, and curvature vector 2 norm to calculate the specific location of damage. The vector norm is used to characterize the distribution characteristics of the difference in strain response caused by structural damage along the axial and circumferential directions, with its peak point corresponding to the coordinate of the damage location.

#### 2.3.1. Strain Vector Norm

The strain vector norms are calculated from the structural strain response difference field of the compartment, which mainly includes two types of strain vector 1 norms (denoted as: *S*1) and strain vector 2 norms (denoted as: *S*2), which are calculated by the following equations, respectively [28]:(24)$$S1=\sum_{i=1}^{N}\left|{\left(\left\{\Delta \varepsilon \right\}\right)}_{i}\right|$$
(25)S2=(∑i=1N|({Δε})i|2)12
where *S*1 and *S*2 are the structural damage discrimination index calculated based on the strain vector 1 and 2, respectively.

The specific damage identification and location discrimination process is as follows: first, invert Equation (23) to obtain the structural strain response difference field of the compartment; second, calculate the two strain vector norm curves along the circumferential and axial directions of the structural strain response difference field of the compartment according to Equation (24) and Equation (25), respectively; thirdly, determine the number of structural damages by the number of peaks along the circumferential and axial curves of the two strain norm curves; lastly, the peak coordinates of the circumferential curve correspond to the structural damages are regarded as circumferential coordinates of the structural damages, while the peak coordinates of the axial curve correspond to the axial curve are regarded as structural damages axial coordinates of the structural damages.

#### 2.3.2. Curvature Vector Norm

The damage coordinate identification metrics calculated based on the curvature vector norms mainly include two types of curvature vector 1 norms (denoted as: *C*1) and curvature vector 2 norms (denoted as: *C*2), which are calculated by the following formulas, respectively [29]:(26)$$C1=\sum_{i=1}^{N}\left|{\left(\left\{\Delta \ddot{\varepsilon }\right\}\right)}_{i}\right|$$
(27)C2=(∑i=1N|({Δε¨})i|2)12
where $\left\{\Delta \ddot{\varepsilon }\right\}$ is the second order derivative of the strain difference vector due to structural damage. *C*1 and *C*2 are the structural damage discrimination index calculated based on the curvature vector 1 and 2, respectively.

The specific damage identification and location discrimination process is as follows: firstly, the second-order derivatives of the strain difference vectors $\left\{\Delta \ddot{\varepsilon }\right\}$ caused by the structural damage are calculated; secondly, the two curvature vector norm curves along the circumferential and axial directions of the structural strain response difference field of the compartment are calculated according to Equations (26) and (27), respectively; thirdly, the number of structural damages is determined by the number of peaks along the two curvature norm curves in the circumferential and axial directions; lastly, the peak coordinate of the circumferential curve of curvature norm curve corresponding to the circumferential direction is considered to be the circumferential coordinate of the structural damages, while that of the axial curve of curvature norm curve corresponding to the axial direction is considered to be the axial coordinate of structural damages.

The spacecraft segment structure damage identification process based on the principle of fiber optic sensor and strain response differential field mutation feature identification is shown in Figure 2.

## 3. Simulation Validation of Segment Damage Recognition Methods

### 3.1. Finite Element Modeling of Damage-Containing Compartment Structure

The spacecraft reinforced segment structure mainly consists of a barrel capsule, axial bars, and circumferential bars, as shown in Figure 3. The structure is made of aluminum alloy, with a density of 2640 kg/m^3^, a modulus of elasticity of 70 GPa, and a Poisson’s ratio of 0.3. Some dimensions of reinforced segment structure are shown in Table 1. The Height of the rib is the distance between the upper surface of the rib and the surface of the cabin, and the thickness is the width of the rib, as shown in Figure 3.

The process of structural finite element calculation is as follows: firstly, a three dimensional model of the spacecraft segment structure is imported into the finite element analysis software Abaqus; secondly, the spacecraft segment structure is meshed with C3D10 cells; thirdly, the axial degrees of freedom of the end faces of the segment structure are constrained; and lastly, a homogeneous pressure load with a magnitude of 91 kPa is applied on the inner wall of the segment, which is used to simulate the internal pressure difference of the segment in the real space environment [30].

Four units in the middle of the reinforced surface of the segment structure are selected as the damage-monitoring area, named Unit A, Unit B, Unit C and Unit D. Each monitoring unit has a circumferential length of 170 mm and an axial length of 170 mm along the cylinder segment. A total of 24 strain extraction points were set up in the four monitoring units, of which 1# to 6# are located in Path 1, which is located in Unit A and Unit B; points 7# to 12# are located in Path 2, which is located in Unit A and Unit B; points 13# to 18# are located in Path 3, which is located in Unit C and Unit D; points 19# to 24# are located in Path 4, which is located in Unit C and Unit D. Path 1, Path 2, Path 3, as well as Path 4, are laid out in parallel with equal spacing along the axial direction of the segment section, as shown in Figure 3. Where, a, b, c, and d represent the four endpoints of the monitoring area.

Due to the impact of space debris on the segment structure, it is easy to cause structural crack damage, so four reinforced grid cells are selected to set up crack damage in the outer Unit A, Unit B, Unit C and Unit D of the segment, and the length of the crack damage is set to be 5 mm, and the depth of the crack is set to be 0.5 mm [31].

### 3.2. Simulation Verification of Damage Region Localization Based on Strain Response Difference

When there is a single crack damage in the Unit A monitoring unit, according to the strain extraction paths from Path 1 to Path 4 in Figure 3, the structural strain response of the segment containing crack damage is obtained, and then according to the inversion of Equation (22), the strain response distribution of the whole monitoring area abcd is obtained, as shown in Figure 4a. From this Figure, it can be seen that the surface strain distribution characteristics of the compartment structure obtained by the inversion of Equation (22) alone cannot intuitively identify the damage location and determine the number of damage.

In order to highlight the sudden change characteristics of the local strain response of the structure caused by the damage, the structural strain response difference field of the compartment segment containing crack damage is obtained by inversion according to Equation (23), as shown in Figure 4b. As can be seen from this figure, through the characterization of the sudden change in the strain response differential field obtained from the reconstruction, it can be found that there is a single crack damage in the compartment segment and the damage is located in the lower right corner of the Unit A region, but it is still not possible to give the exact location coordinates of the crack damage.

When there is double crack damage in the monitoring unit of Unit B and Unit C, the inversion method described in Equation (22) is used to obtain the strain response distribution cloud diagram of the monitoring unit, as shown in Figure 5a. From this Figure, it can be seen that there is a slight change in the strain field on the surface of the structure containing damage, but the strain response distribution cloud diagram is unable to visually characterize the number of damages and the region where the damages are located.

The strain response difference field corresponding to the damage-containing structure is reconstructed using the method described in Equation (23), as shown in Figure 5b. From this Figure, it can be seen that there is a double crack damage in the compartment segment, which is located in the middle region of Unit B and Unit C, respectively. Since the crack damage also causes some abrupt changes in the distribution of the strain response difference in the surrounding region, the precise location of the crack damage cannot be given by this Figure alone.

### 3.3. Simulation Validation of the Accurate Identification Method of Crack Damage Coordinates Based on the Norm Calculation

In order to further realize the precise location of structural crack damage in spacecraft segments, four indicators, including strain vector 1 norm, strain vector 2 norm, curvature vector 1 norm and curvature vector 2 norm, are introduced to characterize the damage response. Based on this, the number of peaks and coordinates of the peaks of the vector norm curves, and the number of structural crack damages are identified and accurately located.

In order to characterize the damage localization results intuitively, and at the same time facilitate the evaluation of the damage identification effect, the four arc-shaped monitoring units Unit A, Unit B, Unit C, and Unit D on the surface of the silo are expanded into four two-dimensional planar monitoring units, which are Unit A’, Unit B’, Unit C’, and Unit D’, respectively, as shown in Figure 6. The circumferential and axial coordinate axes of the 2D planar monitoring units in Figure 6 correspond to the cd arc and ca straight line of the arc-shaped monitoring unit in Figure 3, respectively.

#### 3.3.1. Crack Damage Coordinate Identification Based on Strain Vector Norm Calculation

The middle position of the preset crack is regarded as the damage location in the simulation, and the specific coordinates are (−17 mm, 417 mm), as shown by the crack mark Crack in Figure 6. According to Equations (24) and (25), the strain vector norms under the single crack damage condition shown in Figure 4b are calculated, and the pink rhombus represents the strain vector 1 norm and the green square represents the strain vector 2 norm, as shown in Figure 6. From this Figure, it can be seen that there is a single peak along the circumferential and axial curves of the two strain vector norms, and the peak circumferential coordinates of the strain vector 1 norm and strain vector 2 norm are consistent with the circumferential coordinates of the preset damage, while the peak axial coordinates of both of them are slightly deviated from the axial coordinates of the preset damage.

The peak coordinates of the strain vector 1 parametric curve are calculated from the peak coordinates of the crack damage as (−17 mm, 400 mm), which are shown as the crack identifiers S1 crack in Figure 6. The crack damage coordinates calculated from the peak coordinates of the strain vector 2 norm curve are (−17 mm, 400 mm), as shown in the crack identification S2 crack in Figure 6. The simulation results show that the crack damage coordinates calculated based on the strain vector norm deviate slightly from the axial coordinates of the preset crack damage.

The double crack damage locations are preset as (−85 mm, 315 mm) and (85 mm, 485 mm) in the simulation, as shown in the crack identification Crack in Figure 7. According to Equations (24) and (25), the strain vector norms under the double crack damage condition shown in Figure 5b are calculated, and the pink rhombus represents the strain vector 1 norm and the green square represents the strain vector 2 norm, as shown in Figure 7. From this figure, it can be seen that there are double peaks along the circumferential and axial curves, and the peak circumferential and axial coordinates of the strain vector 1 norm deviate slightly from the preset damage circumferential and axial coordinates, and the same phenomenon exists in the strain vector 2 norm.

The damage coordinates of the double cracks calculated by the strain vector 1 norm are (−68 mm, 315 mm) and (85 mm, 502 mm), which are shown as the crack marks S1 crack in Figure 7. The double crack damage coordinates calculated from the strain vector 2 norm are (−85 mm, 332 mm), and (68 mm, 485 mm), respectively, as shown in the crack identification S2 crack in Figure 7. The simulation results show that there is a certain deviation in the accurate identification of the double crack damage location based on the strain vector norm.

#### 3.3.2. Crack Damage Coordinate Identification Based on Curvature Vector Norm

The crack damage location is preset as (−17 mm, 417 mm) in the simulation, as shown by the crack identification Crack in Figure 8. The curvature vector norms under the single crack damage condition shown in Figure 4b are calculated according to Equations (26) and (27), with the red circles representing the curvature vector 1 norm and the blue triangles representing the curvature vector 2 norm, as shown in Figure 8.

From this figure, it can be seen that there is a single peak of the two curvature vector norm curves along the circumferential and axial directions. The crack damage coordinates calculated from the curvature vector 1 norm are (−17 mm, 400 mm), and the axial coordinates of the peak of the curvature vector 1 norm curve are consistent with the axial coordinates of the preset damage, while the circumferential coordinates of the peak are slightly deviated from the circumferential coordinates of the preset damage as shown by the crack markings C1 crack in Figure 8. In contrast, the crack damage coordinates calculated from the curvature vector 2 norm are (−17 mm, 417 mm), and the axial and circumferential coordinates of the peak curvature vector 2 norm curve are consistent with the axial and circumferential coordinates of the preset damage, respectively, as shown by the crack identifier C2 crack in Figure 8.

The preset crack damage locations in the simulation are (−85 mm, 315 mm) and (85 mm, 485 mm), as shown by the crack mark Crack in Figure 9. According to Equations (26) and (27), the curvature vector norms under the double crack damage condition shown in Figure 5b are calculated, in which the red hollow circle represents the curvature vector 1 norm and the blue hollow triangle represents the curvature vector 2 norm, as shown in Figure 9. From this figure, it can be seen that there are double peaks along the two strain vector norm curves in the circumferential and axial directions.

The double crack damage coordinates calculated by the curvature vector 1 norm are (−68 mm, 315 mm) and (85 mm, 485 mm), respectively. The peak coordinates of the curvature vector 1 norm curve along the circumferential direction deviate slightly from the preset damage circumferential coordinates, while the axial peak coordinates are in agreement with the pre-set damage axial coordinates, as shown by the crack marker C1 crack in Figure 9.

The double crack damage coordinates calculated by the curvature vector 2 norm are (−85 mm, 315 mm) and (85 mm, 485 mm), and the peak circumferential and axial coordinates of the curvature vector 2 norm coincide with the preset circumferential and axial coordinates of the damage, as shown in the crack identifier C2 crack in Figure 9, respectively. The simulation results show that the curvature vector 2 norm is optimal in the accurate identification of double crack damage location.

#### 3.3.3. Comparison of Crag Factors Corresponding to Different Norms

According to the simulation results, it is found that there is a difference in the cliff factor of the peak curves of the four vector norms corresponding to the same damage, and the larger the cliff factor, the higher the spatial resolution of the norm for the damage location [32]. In order to compare the degree of steepness of the peak curves of the four norms, the cliff factors corresponding to single and double damage are calculated, respectively, as shown in Figure 10 and Figure 11.

As can be seen from Figure 10, under the single crack damage condition, the cliff factors corresponding to the peak curves of the norms along the circumferential and axial directions increase sequentially, which indicates that the spatial resolution of the damage corresponding to the four vector norms also increases gradually.

From Figure 11, it can be seen that the norm cliff factor corresponding to the double crack damage shows the same increasing law as that of the single crack damage, and the curvature vector 2 norm cliff factor is the largest in this case, i.e., the effect of the damage peak mutation it characterizes is more obvious, and the selection of this norm can help improve the accuracy of the crack damage location identification.

## 4. Experimental System Construction

### 4.1. Principle of Fiber Grating Sensor

Fiber Bragg grating sensor is a wavelength modulated sensing element that modulates the grating center wavelength by external physical quantities to be measured, such as temperature and strain changes, to obtain sensing information [33]. The fiber grating device sensing principle is shown in Figure 12.

If the fiber grating sensor is subjected to external axial loading only in a constant temperature environment, the period and effective refractive index of its grating region change, which leads to a shift in its central wavelength, and its strain expression is [34]:(28)$$\varepsilon =\frac{1}{\left(1-P_{e}\right)}\cdot \frac{\Delta \lambda_{B}}{\lambda_{B}}=\frac{1}{\left(1-P_{e}\right)}\cdot \left(\frac{\Delta n_{eff}}{n_{eff}}+\frac{\Delta \Lambda }{\Lambda }\right)$$
where *P_e_* denotes the elasticity coefficient, *λ_B_* is the center wavelength, Δ*λ_B_* is the center wavelength shift, *n_eff_* is the effective refractive index, Δ*n_eff_* is the finite refractive index change, Λ is the grid period, and △Λ is the change in the grid period.

Considering the cross-sensitivity of temperature and strain of optical fiber sensors in the actual service environment, temperature compensation technology can be used to solve the problem. The optical fiber temperature sensor is connected in series in the monitoring system to compensate the interference of temperature to the actual strain measurement.

### 4.2. Fiber Grating Sensor Layout

A fiber grating sensor string with six FBGs is deployed at 1/3 and 2/3 of the axial direction (*Y*-axis) of the segment structure inside the monitoring units of Unit A, Unit B, Unit C, and Unit D, respectively, to obtain the strain response characteristics of the measurement point locations. A schematic diagram of the spacecraft segment structure damage monitoring area and the fiber grating monitoring network is shown in Figure 13.

Considering the processing efficiency and experimental cost factors, and in order to ensure that the strain response characteristics caused by the preset damage approximate the real damage of the spacecraft structure, this paper adopts the induced simulated damage to approximate the actual damage [15,35]. Since the damage in the segment structure will lead to the reduction of structural stiffness, specific weights are placed to simulate the crack damage, as shown in Figure 13. The location and amount of crack damage is simulated by changing the area and amount of weights placed.

### 4.3. Construction of Structural Damage Monitoring System for Segments

The structure damage monitoring experiment system based on fiber grating sensors mainly includes: typical segment structure, MOI SI155 interrogator, solid support stand, hydraulic jack and computer, etc., as shown in Figure 14.

A uniform load-loading device is constructed inside the segment structure to simulate the working condition of uniform pressure difference between the inside and outside of the segment during the on orbit operation of a real spacecraft. The bottom end of the jack is placed on the support frame, and the top end is kept in contact with the arc-shaped pad. The arc-shaped plate converts the concentrated force exerted by the jack into a uniform load, which is applied to the monitoring unit to simulate the real load condition of the shell of the actual segment in orbit, as shown in Figure 15.

## 5. Experimental Results and Discussion

In order to visually characterize the damage localization results, the four arc monitoring units Unit A, Unit B, Unit C and Unit D on the silo surface are expanded into four 2D planar monitoring units, Unit A’, Unit B’, Unit C’ and Unit D’, respectively, as shown in Figure 16. The circumferential and axial coordinate axes of the 2D planar monitoring unit in Figure 16b correspond to the arcs and *bd* lines of the arc monitoring unit in Figure 16a, respectively.

Where, a, b, c, and d represent the four endpoints of the three-dimensional monitoring area respectively, and a′, b′, c′, and d′ represent the four endpoints of the expanded two-dimensional monitoring area respectively.

### 5.1. Evaluation of the Effectiveness of Single-Damage Identification

#### 5.1.1. Single-Damage Region Identification Based on Strain Response Difference Field

The single-damage condition is preset in the Unit C monitoring unit, and the strain response difference field in the monitoring area is obtained by inversion according to Equation (23), as shown in Figure 16. From this Figure, it can be seen that there is an obvious strain response difference mutation inside the Unit C monitoring unit, and the actual damage location is located inside the red circular spot.

#### 5.1.2. Single-Damage Coordinate Identification Based on Different Norm Calculations

In order to compare the damage coordinates identification effect corresponding to the four vector norms, the working condition shown in Figure 16b is selected. In the experiment, the coordinates of the simulated damage location are set at (−34 mm, −68 mm), as shown in the yellow circular damage marker Damage in Figure 17. According to Equations (24) and (25), the strain vector 1 norm and strain vector 2 norm corresponding to the working condition shown in Figure 16b are calculated, as shown in Figure 17. From this figure, it can be seen that there is one peak value in both strain vector norm curves along the circumferential and axial directions of the nacelle section, indicating the existence of a single damage, which is consistent with the actual structure containing the number of damages. Meanwhile, according to the circumferential and axial coordinates of the peaks of the strain vector norm curves, the exact location of the damage can be determined.

The single-damage coordinates are calculated from the peak coordinates of the strain vector 1 parametric curve as (−17 mm, −51 mm), as shown in the pink circular damage marker S1-Damage in Figure 17a. According to the peak coordinates of the strain vector 2 norm curve, the single-damage coordinates are calculated as (−17 mm, −68 mm), as shown in the green circular damage mark S2-Damage in Figure 17b. All of the above identification results have some deviation from the actual damage location.

According to Equations (26) and (27), the curvature vector 1 norm and curvature vector 2 norm corresponding to the working condition shown in Figure 16b are calculated, as shown in Figure 18. From this Figure, it can be seen that both curvature vector norm curves along the circumferential and axial directions of the nacelle section have one peak, indicating the existence of single damage, which is consistent with the actual structure containing the number of damages. Meanwhile, according to the circumferential and axial coordinates of the peaks of the curvature vector norm curves, the exact location of the damage can be determined.

According to the peak coordinates of the curvature vector 1 norm curve, the single-damage coordinates are calculated as (−34 mm, −85 mm), which is closer to the actual damage location, as shown in the red circular damage mark C1-Damage in Figure 18a. According to the peak coordinates of the curvature vector 2 norm curve, the single-damage coordinates are calculated as (−34 mm, −68 mm), which is consistent with the actual damage location. Under this condition, the curvature vector 2 norm damage is optimized, as shown by the blue circular damage mark C2-Damage in Figure 18b.

In order to compare the cliff of the peaks of the four parametric curves, the parametric gradient factors corresponding to the working conditions shown in Figure 16 are calculated, as shown in Figure 19. From this figure, it can be seen that the curvature norm cliff factor is larger than the strain norm cliff factor, and the curvature vector 2 norm cliff factor is significantly larger than the other three vector norms, which indicates that this norm has a higher spatial resolution for the damage location, and it helps to improve the accuracy of the damage coordinate identification.

#### 5.1.3. Evaluation of the Effectiveness of Single-Damage Identification

A single damage is prefabricated at different locations in the area, and the coordinates of the damage location are calculated based on the curvature vector 2 norm, with the yellow solid circle representing the actual damage location and the blue solid circle representing the damage coordinate calculation results, as shown in Figure 20. From this figure, it can be seen that the damage location calculated based on the peak coordinates of the curvature vector 2 norm curve matches well with the actual damage location.

In order to quantitatively evaluate the damage localization effects corresponding to the four different vector norms, the absolute error, the average error, and the root mean square error of damage location identification are calculated according to Equations (29)–(31) using the preset damage coordinates as the reference.
(29)$$AE(i)=\sqrt{{\left(x(i)-x^{Ref}(i)\right)}^{2}+{\left(y(i)-y^{Ref}(i)\right)}^{2}}$$
(30)$$MRE=\frac{\sum_{i=0}^{n}AE(i)}{n}$$
(31)$$RMSE=\sqrt[{2}]{\frac{\sum_{i=0}^{n}{\left(AE(i)-MRE\right)}^{2}}{n}}$$
where *x^Ref^*(*i*) and *y^Ref^*(*i*) are the actual circumferential and axial coordinates of the *i*th damage location, respectively, *x*(*i*) and *y*(*i*) are the computed results of the circumferential and axial coordinates of the *i*th damage location, respectively, *AE*(*i*) is the absolute error of the identification result of the *i*th damage location, *MRE* is the average error of the identification results of the location of the n damage points, and *RMSE* is the root-mean-square error of the identification result of the n damage locations.

The localization errors of the four vector norms for each damage location in Figure 20 are calculated separately, as shown in Table 2. From this table, it can be seen that the average damage localization error and the root mean square error calculated from strain vector 1 norm, strain vector 2 norm, curvature vector 1 norm and curvature vector 2 norm are in decreasing order. The average error of single-damage localization calculated by the curvature vector 2 norm C2 is 2.58 mm and the root-mean-square error is 1.98 mm, which is better than the other three vector norms. In Figure 19, it is shown that the curvature vector 2 norm C2 has the highest cliff factor, which corresponds to the best damage identification accuracy of this norm in Table 2, and verifies that this norm has a high damage spatial resolution.

### 5.2. Effectiveness of Double Crack Damage Identification

#### 5.2.1. Damage Region Identification Based on Strain Response Difference Field

In the Unit C and Unit D monitoring units, the double-damage condition is preset, and the strain response difference field of the whole monitoring region is obtained by inversion according to Equation (23), as shown in Figure 21. From this figure, it can be seen that there are two obvious strain response difference mutation regions in the monitoring unit, which are located in the middle region of the Unit C monitoring unit and the upper left region of the Unit D monitoring unit, which indicates that two damages have occurred in the above region.

Where, a, b, c, and d represent the four endpoints of the three-dimensional monitoring area respectively, and a′, b′, c′, and d′ represent the four endpoints of the expanded two-dimensional monitoring area respectively.

#### 5.2.2. Damage Coordinate Identification Based on Norm Calculation

In order to compare the effect of accurate damage coordinate identification based on the strain vector norm calculation, the strain vector 1 norm and the strain vector 2 norm are calculated as shown in Figure 21 for the double-damage condition, as shown in Figure 22. In the experiment, the coordinates of the simulated double-damage locations are (−102 mm, −102 mm) and (34 mm, −34 mm), as shown in the yellow circular damage mark Damage in Figure 22.

The double-damage coordinates calculated from the strain vector 1 norm are (−85 mm, −85 mm), and (34 mm, −17 mm), as shown by the pink circular damage marker S1-Damage in Figure 22a. The double-damage coordinates calculated from the strain vector 2 norm are (−102 mm, −85 mm), and (34 mm, −17 mm), respectively, as shown in the green circular damage mark S2-Damage in Figure 22b. The results show that there is some error between the damage localization coordinates calculated based on the strain vector norm and the actual damage location.

In order to compare the effect of accurate damage coordinate identification based on the curvature vector norm calculation, the curvature vector 1 norm and curvature vector 2 norm are calculated under the double-damage condition shown in Figure 21, as shown in Figure 23. In the experiment, the coordinates of the simulated double-damage locations are set as (−102 mm, −102 mm) and (34 mm, −34 mm), as shown in the yellow circular damage mark Damage in Figure 23.

According to the curvature vector 1 norm, the double-damage coordinates are (−102 mm, −102 mm), and (51 mm, −34 mm), which are close to the actual damage location, as shown in the red circular damage mark C1-Damage in Figure 23a. The double-damage coordinates calculated by the curvature vector 2 norm are (−102 mm, −102 mm), and (34 mm, −34 mm), which are consistent with the actual damage location, i.e., the curvature vector 2 norm damage judgment is optimal in this condition, as shown in the blue circular damage mark C2-Damage in Figure 23b.

The cliff factors of the norm curves corresponding to the working conditions shown in Figure 21 are calculated and shown in Figure 24. From this figure, it can be seen that the cliff factors of the four norm curves of strain vector 1 norm, strain vector 2 norm, curvature vector 1 norm, and curvature vector 2 norm show an obvious increasing trend, which makes the spatial discrimination accuracy of the four norms for the damage location also increase step by step. According to the identification results shown in Figure 22 and Figure 23, the above law can also be verified. The curvature vector 2 norm has the highest cliff factor, which makes it have higher damage spatial resolution, and thus obtains the optimal damage coordinate identification.

#### 5.2.3. Evaluation of the Effectiveness of Double-Damage Identification

The double damage is prefabricated at different locations in the area, and the coordinates of the damage location are calculated based on the curvature vector 2 norm, with the yellow solid circle representing the actual damage location and the blue solid circle representing the damage coordinate calculation results, as shown in Figure 25. From this figure, it can be seen that the damage location calculated based on the peak coordinates of the curvature vector 2 norm curve matches well with the actual damage location.

The localization errors corresponding to strain vector 1 norm S1, strain vector 2 norm S2, curvature vector 1 norm C1, and curvature vector 2 norm C2 for each damage location in Figure 25 are calculated, respectively, as shown in Table 3.

From Table 3, it can be seen that the average localization error and the root mean square error obtained from the calculation based on the curvature norm 2 norm C2 are smaller than the other three norms. Strain vector 1 norm S1, strain vector 2 norm S2, curvature vector 1 norm C1, curvature vector 2 norm C2 and the other four vectors’ norm damage identification accuracy increases step by step, consistent with the change rule of norm curve crag factor in Figure 24.

## 6. Conclusions

Considering the structural health monitoring needs of spacecraft segments, this paper proposes a structural damage identification method based on fiber optic sensors and strain response difference feature calculation. The research results can be used in the field of spacecraft structural health monitoring and digital twin, and have good engineering applicability.

(1) A damage region identification method based on the principle of strain response difference field inversion is developed, and the strain response difference features corresponding to the damage at different locations are obtained by equating the structural damage to the concentrated additional load on the defect free structure. Without the need of a priori information of structural damage, the method realizes the determination of the crack damage region, which solves the problem of not being able to identify the number and location of damages in the structures containing crack damages based on the strain distribution characteristics alone.

(2) Based on strain vector 1 norm, strain vector 2 norm, curvature vector 1 norm, and curvature vector 2 norm, a damage location coordinate identification method is proposed, which forms a set of precise identification process of damage location of the compartment structure with “the region” followed by “the coordinates”. The results show that the localization error based on the curvature norm is smaller than that based on the strain norm, and the accuracy of the four vector norms for damage identification increases step by step.

(3) The crag factor for evaluating the mutation characteristics of different vector norm curves is proposed, and a spacecraft segment structure damage monitoring system based on fiber grating sensors is constructed. The experimental results show that the curvature vector 2 norm curves have the highest cliff factor, which results in high damage spatial resolution and optimal damage coordinate identification.

(4) It is worth pointing out that the algorithm described in this paper cleverly uses the difference between the internal and external pressures of spacecraft segments in the real space environment to identify structural damage, obtains the characteristic distribution of the damage response that is equivalent to the global actual measurements, solves the difficult problem of external excitation that needs to be applied by the conventional identification method, and realizes the on-line monitoring of structural damage of spacecraft.

It is worth pointing out that the fiber grating sensor also has some shortcomings, such as its low frequency response, so it is not suitable for collecting high frequency information generated by transient damage like piezoelectric sensors. In addition, optical fibers are passive devices and cannot be used for active health monitoring similar to piezoelectric devices. Therefore, it is necessary to develop a new optical fiber monitoring technology with high frequency sampling characteristics and an active and passive structure health monitoring technology based on the combination mode of optical fiber and piezoelectric.

## Figures and Tables

**Figure 1 sensors-23-08822-f001:**
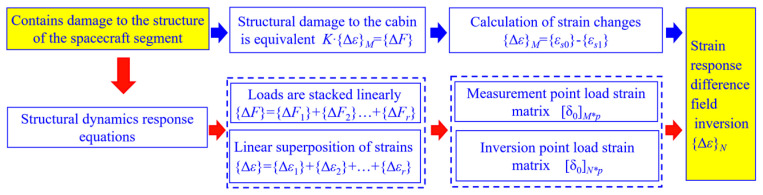
Flow of strain response difference field inversion algorithm for damage-containing compartment structure.

**Figure 2 sensors-23-08822-f002:**
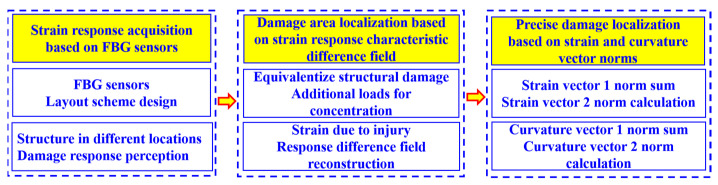
Flowchart of the method for structural damage monitoring and location identification of spacecraft segments.

**Figure 3 sensors-23-08822-f003:**
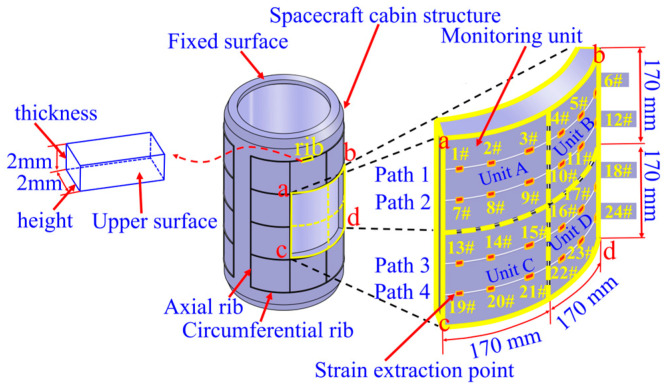
Structure and strain extraction path of a typical spacecraft reinforced segment.

**Figure 4 sensors-23-08822-f004:**
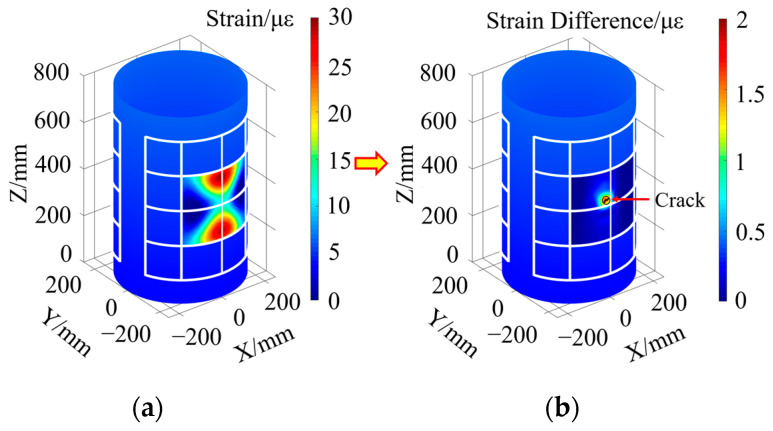
Strain response distribution cloud and strain response difference field in the presence of single crack damage in Unit A monitoring unit: (**a**) strain response distribution cloud; (**b**) strain response characteristic difference field.

**Figure 5 sensors-23-08822-f005:**
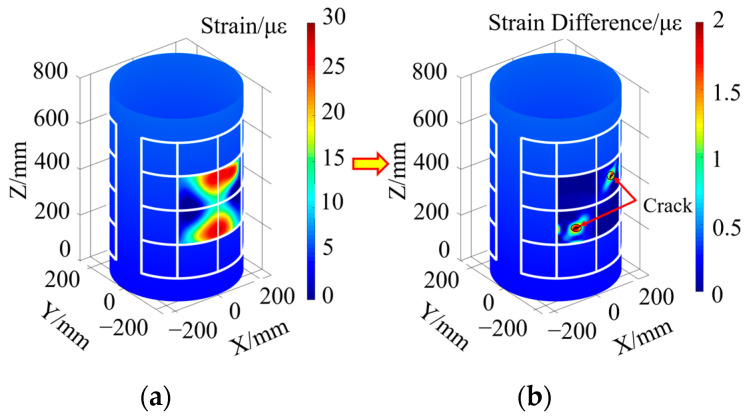
Cloud diagram of strain response distribution and strain response characteristic difference field in the presence of double crack damage in Unit B and Unit C monitoring unit: (**a**) strain response distribution cloud; (**b**) strain response characteristic difference field.

**Figure 6 sensors-23-08822-f006:**
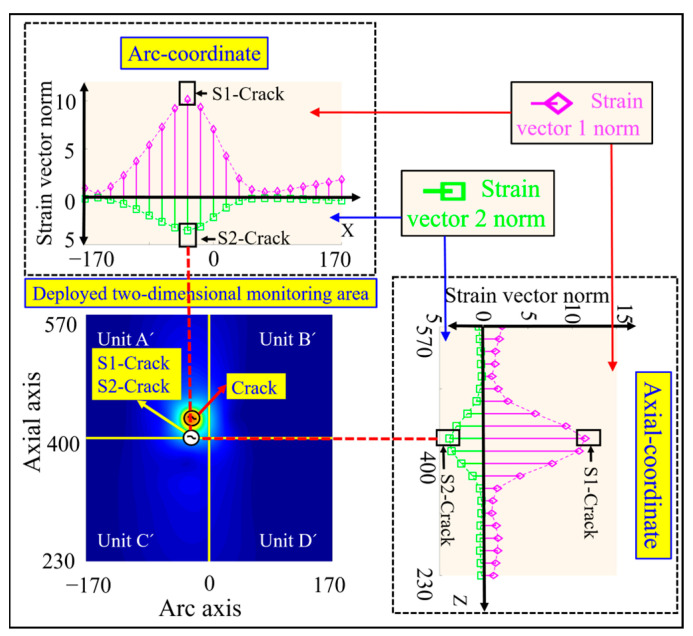
Identification of single crack damage coordinates calculated based on strain vector 1 norm, and strain vector 2 norm.

**Figure 7 sensors-23-08822-f007:**
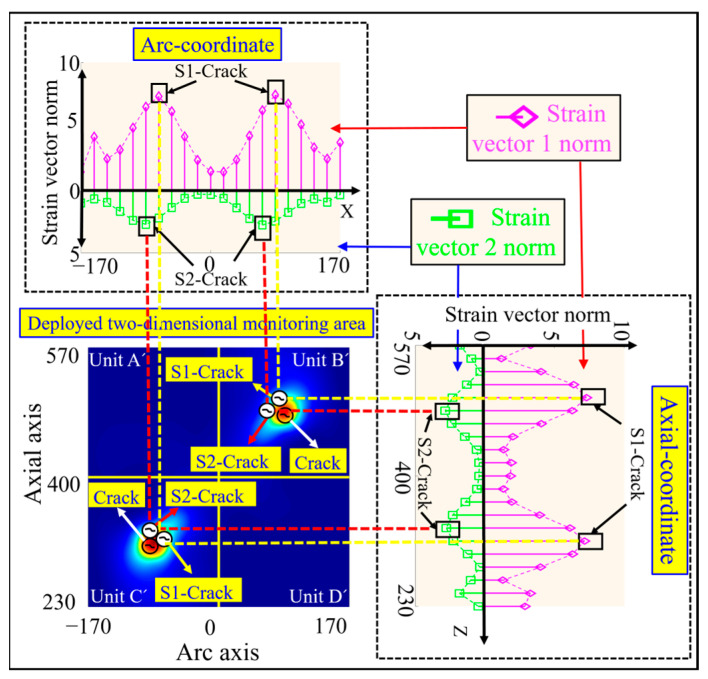
Results of double crack damage coordinate identification based on strain vector 1 norm and strain vector 2 norm calculation.

**Figure 8 sensors-23-08822-f008:**
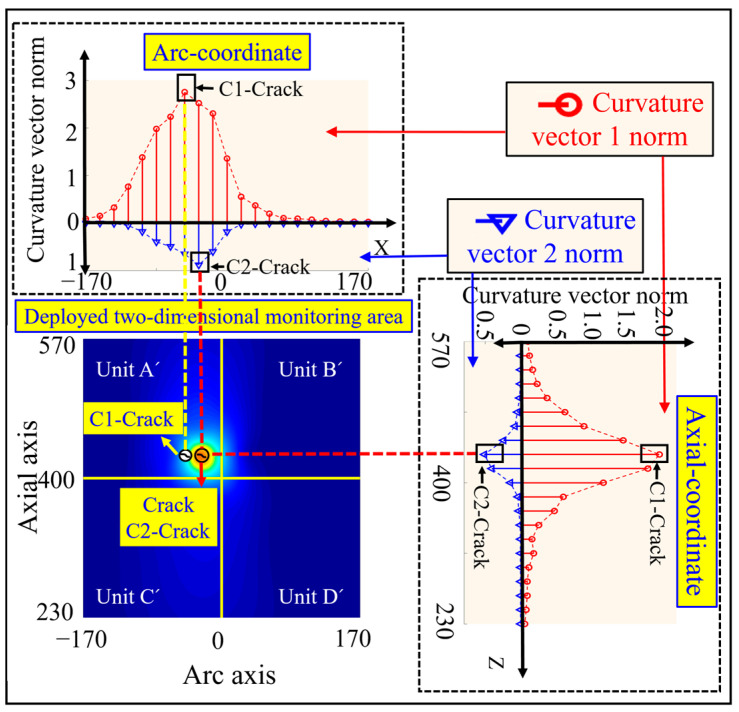
Results of single crack damage coordinate identification based on curvature vector 1 norm and curvature vector 2 norm calculation.

**Figure 9 sensors-23-08822-f009:**
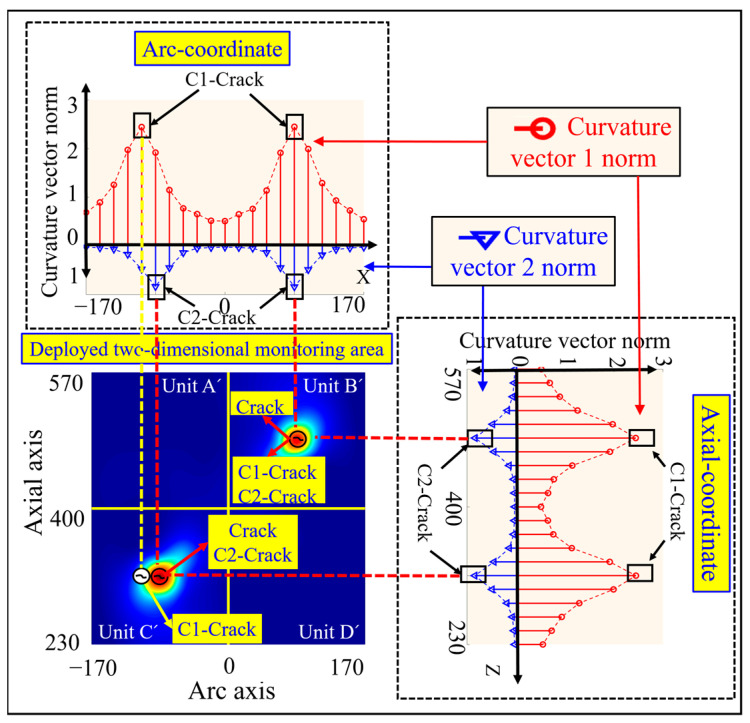
Results of double crack damage coordinate identification based on curvature vector 1 norm and curvature vector 2 norm calculation.

**Figure 10 sensors-23-08822-f010:**
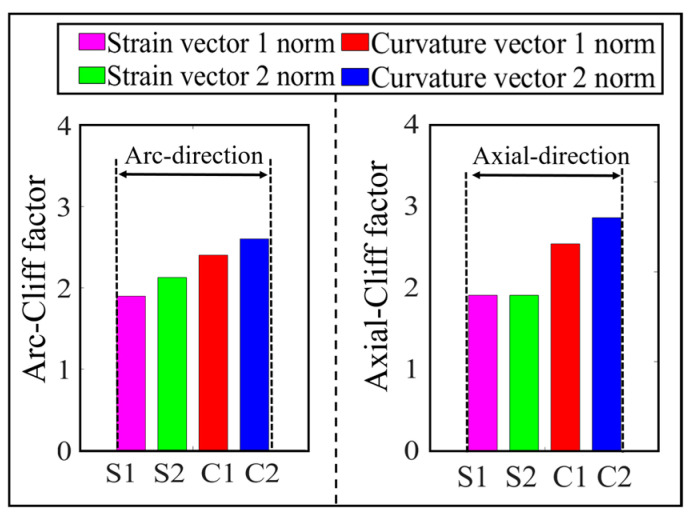
Comparison of 4 vector norm cliff factors corresponding to single damage.

**Figure 11 sensors-23-08822-f011:**
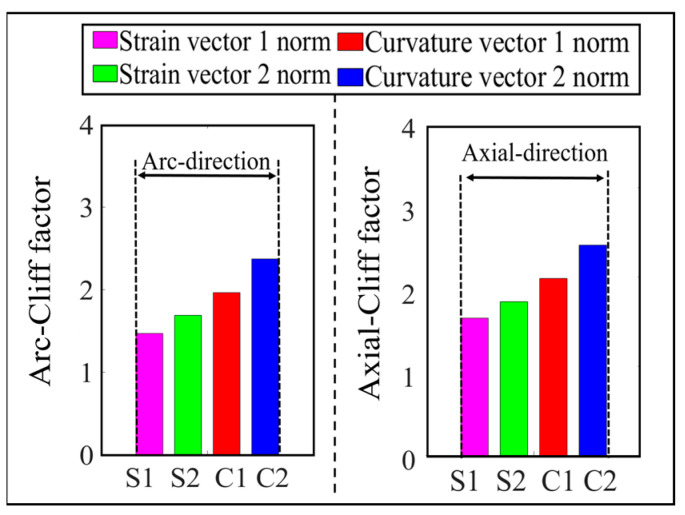
Comparison of 4 vector norm cliff factors corresponding to double damage.

**Figure 12 sensors-23-08822-f012:**
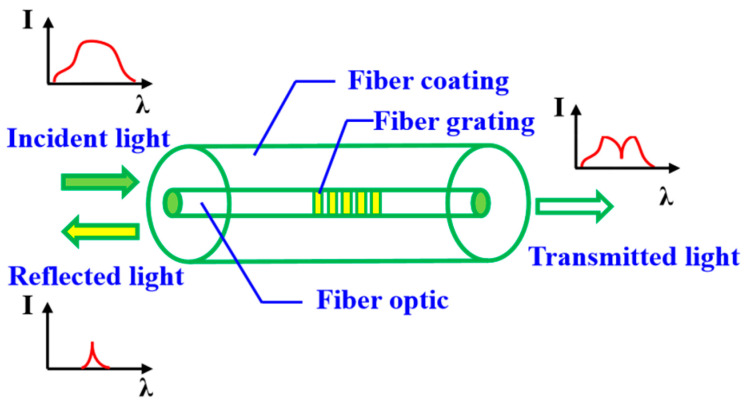
Principle of fiber grating sensor.

**Figure 13 sensors-23-08822-f013:**
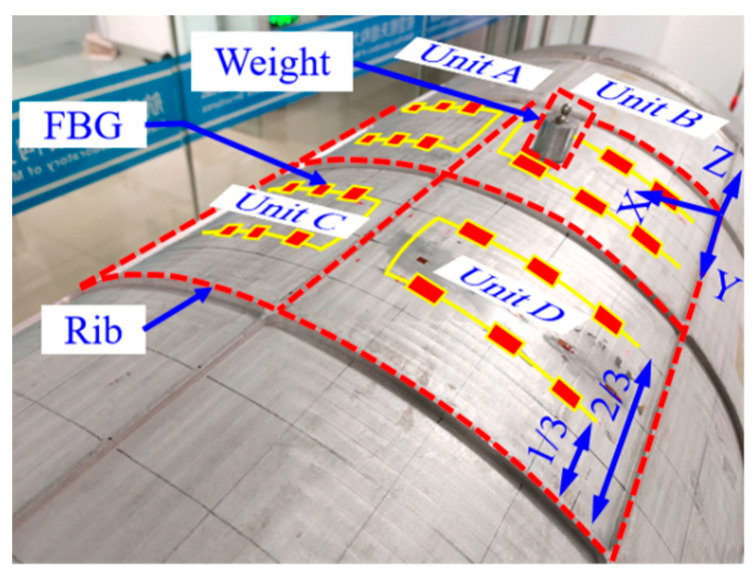
Fiber grating sensor layout and simulated crack damage.

**Figure 14 sensors-23-08822-f014:**
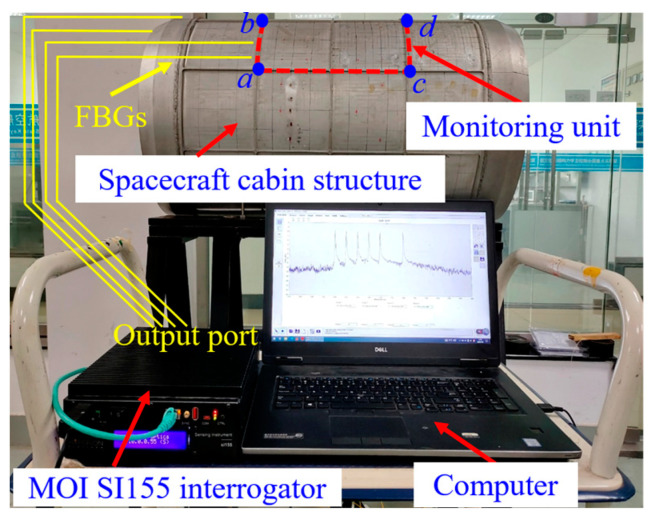
Spacecraft segment structural damage monitoring experiment system.

**Figure 15 sensors-23-08822-f015:**
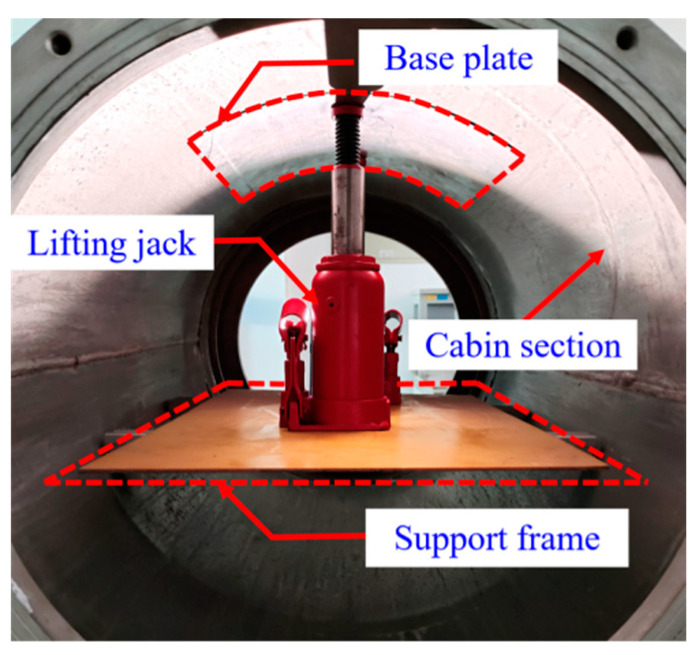
Uniform load application form.

**Figure 16 sensors-23-08822-f016:**
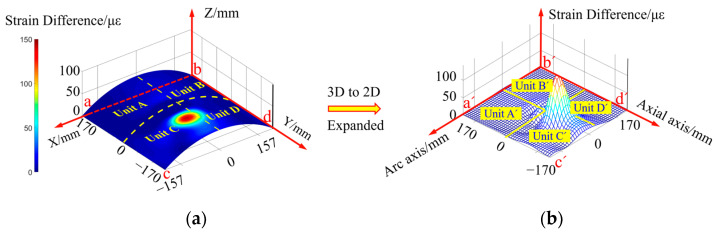
Distribution of strain response difference corresponding to the presence of single damage in Unit C monitoring unit: (**a**) three-dimensional arc monitoring area; (**b**) expanded two-dimensional monitoring area.

**Figure 17 sensors-23-08822-f017:**
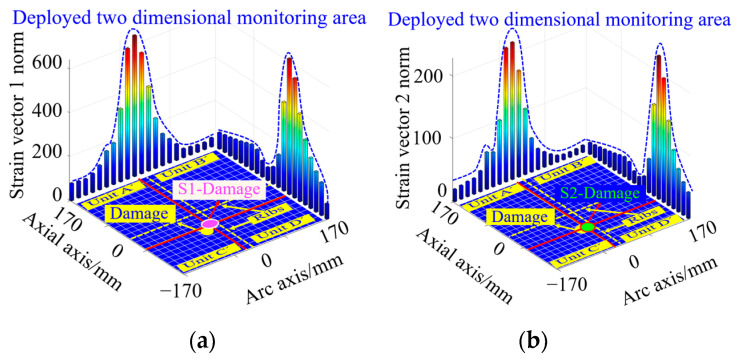
Results of single-damage coordinate identification based on strain vector norm calculation: (**a**) strain vector 1 norm damage coordinate identification result; (**b**) strain vector 2 norm damage coordinate identification result.

**Figure 18 sensors-23-08822-f018:**
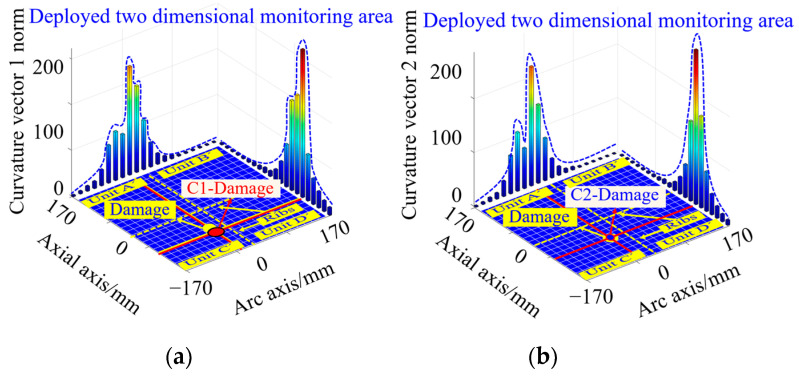
Results of single-damage coordinate identification based on curvature vector norm calculation: (**a**) curvature vector 1 norm damage coordinate identification result; (**b**) curvature vector 2 norm damage coordinate identification result.

**Figure 19 sensors-23-08822-f019:**
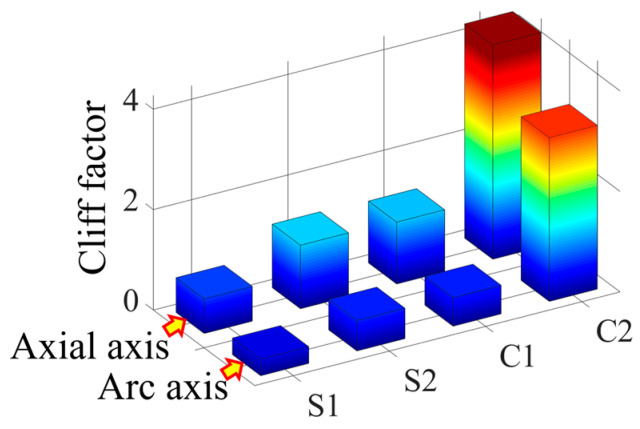
Comparison of single-damage cliff factors obtained from 4 norm calculations.

**Figure 20 sensors-23-08822-f020:**
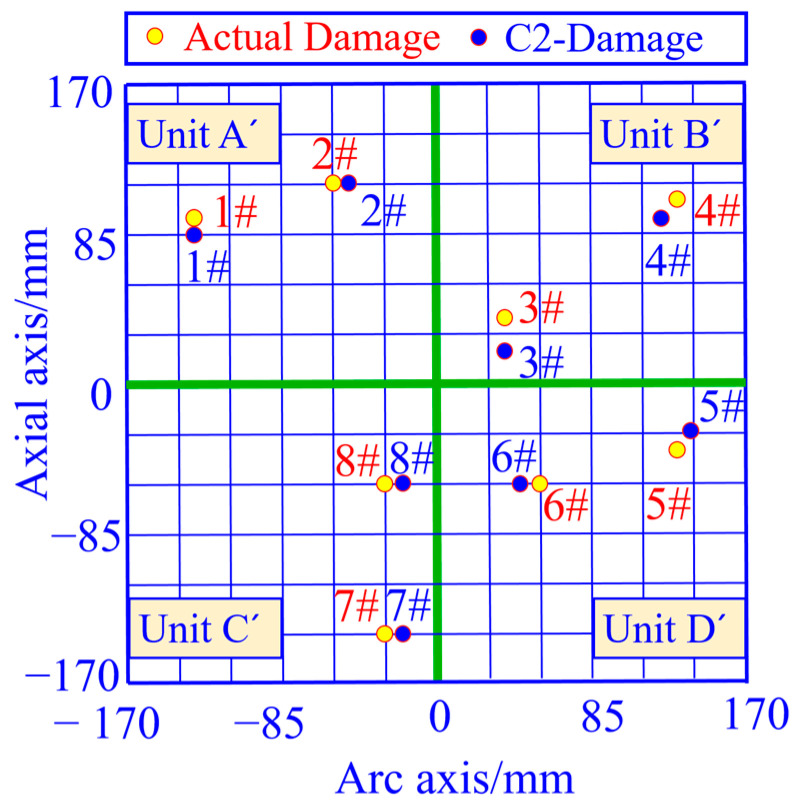
Single-damage localization results of the unfolded 2D planar monitoring region based on the curvature vector 2 norm calculation.

**Figure 21 sensors-23-08822-f021:**
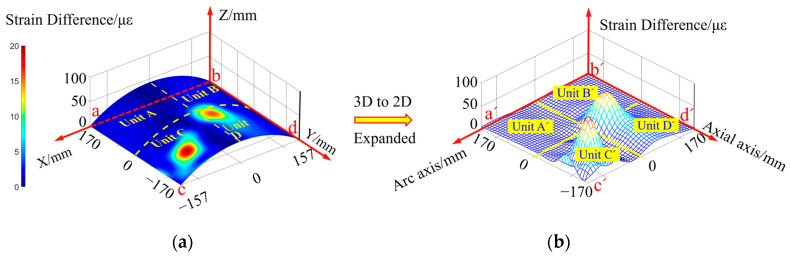
Distribution of strain response difference corresponding to the presence of double damage in Unit C and Unit D monitoring units: (**a**) three-dimensional arc monitoring area; (**b**) expanded two-dimensional monitoring area.

**Figure 22 sensors-23-08822-f022:**
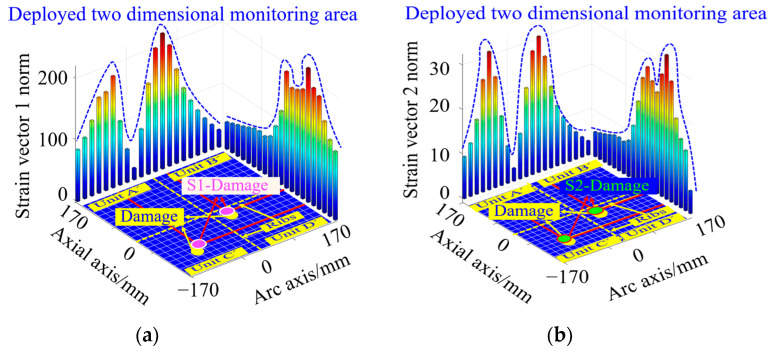
Results of double-damage coordinate identification based on strain vector norm calculation: (**a**) strain vector 1 norm damage coordinate identification result; (**b**) strain vector 2 norm damage coordinate identification result.

**Figure 23 sensors-23-08822-f023:**
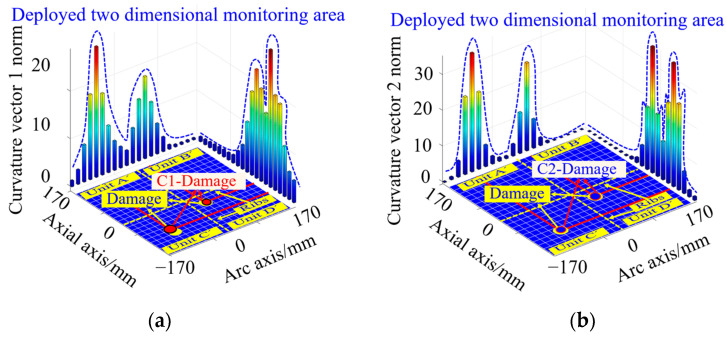
Results of double-damage coordinate identification based on curvature vector norm calculation: (**a**) curvature vector 1 norm damage coordinate identification result; (**b**) curvature vector 2 norm damage coordinate identification result.

**Figure 24 sensors-23-08822-f024:**
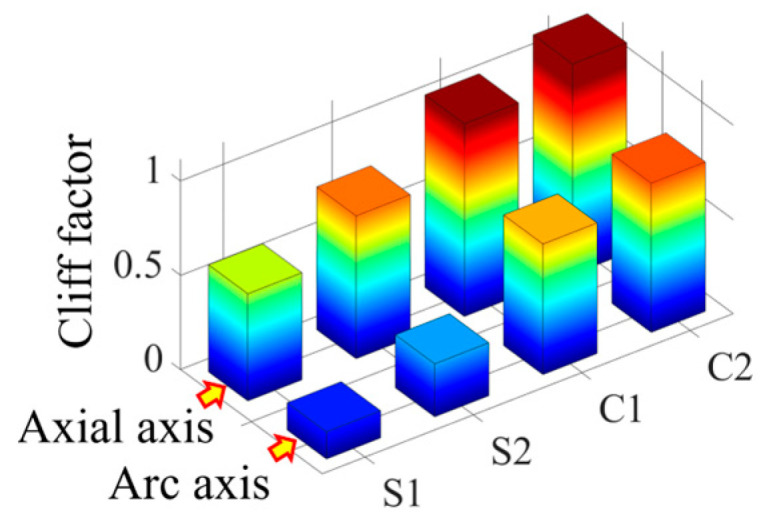
Comparison of double-damage cliff factors obtained from 4 norm calculations.

**Figure 25 sensors-23-08822-f025:**
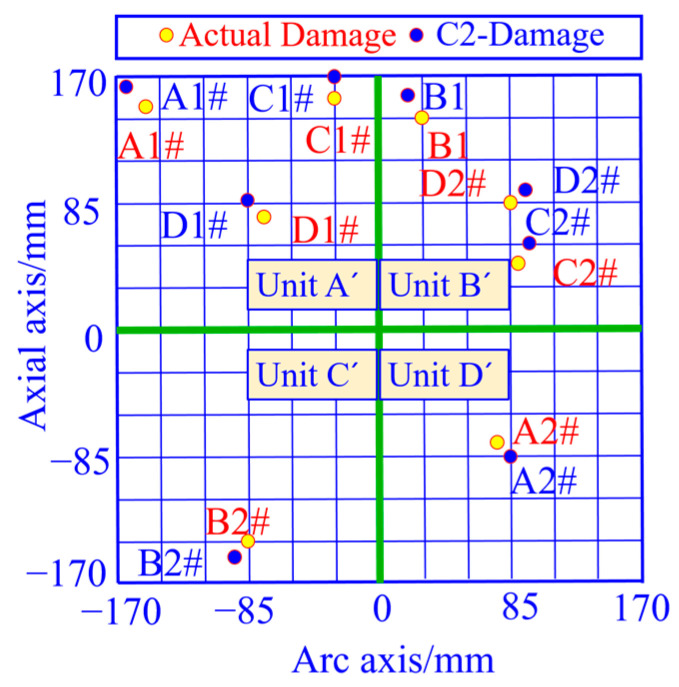
Double-damage localization results based on curvature vector 2 norm computation in the unfolded 2D plane monitoring area.

**Table 1 sensors-23-08822-t001:** Dimensions of structural parts of spacecraft reinforced segments.

	Overall Dimensions/mm		
capsule	diameter	height	wall thickness
	500	800	8
rib	height		thickness
		2	2

**Table 2 sensors-23-08822-t002:** Single-damage localization errors calculated based on four vector norms.

Damage Point Number	Actual Coordinate Value (mm)	Strain Vector Norm Localization Error (mm)		Curvature Vector Norm Localization Error (mm)	
		*S*1	*S*2	*C*1	*C*2
1#	(−130, 85)	9.51	9.51	7.00	4.03
2#	(−56, 112)	2.70	1.11	0.50	0.50
3#	(35, 30)	6.10	6.08	5.85	5.60
4#	(130, 110)	10.92	8.86	6.04	5.39
5#	(135, −30)	7.11	6.80	4.30	2.00
6#	(56, −56)	4.47	2.92	2.06	1.50
7#	(−28, −140)	10.01	8.51	2.55	1.12
8#	(−28, −56)	11.28	8.25	2.00	0.50
*MRE*		7.76	6.50	3.79	2.58
*RMSE*		2.96	2.82	2.19	1.98

**Table 3 sensors-23-08822-t003:** Double-damage localization errors calculated based on four vector norms.

Damage Point Number	Actual Coordinate Value (mm)	Strain Vector Norm Localization Error (mm)		Curvature Vector Norm Localization Error (mm)	
		*S*1	*S*2	*C*1	*C*2
A1#	(−150, 150)	8.07	7.48	7.11	5.41
A2#	(80, −80)	7.21	6.4	6.36	5.7
B1#	(−84, −140)	11.01	4.03	2.12	2.12
B2#	(28, 140)	9	5.5	2.58	1.5
C1#	(−28, 145)	10.93	8.07	5.83	5.83
C2#	(85, 50)	9.72	4.96	4.24	4.24
D1#	(−70, 55)	9.51	8.51	6.51	6.01
D2#	(84, 84)	3.54	2.55	2.55	2.1
*MRE*		8.62	5.93	4.66	4.11
*RMSE*		2.27	1.93	1.90	1.78

## Data Availability

The data presented in this study are not publicly available at this time but may be obtained upon reasonable request from the authors.

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
