# Peer review of "Spacecraft Segment Damage Identification Method Based on Fiber Optic Strain Difference Field Reconstruction and Norm Calculation"

_sensors, 2023, doi:10.3390/s23218822_

Round 1
Reviewer 1 Report
Comments and Suggestions for Authors
Comment attached.

The comment includes a few examples of places that needed language revision. Please revise the manuscript thoroughly.
Reviewer 2 Report
Comments and Suggestions for Authors
Due to the obsolescence of long-used space equipment and orbital stations, this work is quite relevant and is of interest to researchers who are readers of Sensors. I believe that this manuscript deserves publication in this respected journal, but I would suggest a few minor edits:
1. I would like to suggest that authors improve their explanation of the reasons why they choose FBGs. The literature review mentions distributed fiber optic sensors, but does not explain why this option is unacceptable. One of the links refers to quasi-distributed sensors. Minor breakdowns can occur not only in those locations where Bragg gratings are applied, but also in other places. Frequency domain reflectometers have achieved significant results in dynamic range and sensitivity [10.3390/s23156863], [10.1134/S0020441223050172], [10.3390/s23125515], could you be so kind as to explain why this method is unacceptable for your study.
2. The method by which probing is performed is not well described. Gratings can be interrogated in different ways, for which there are quite a few approaches. If a commercial interrogator was used, the brand must be specified. It is advisable to provide a figure (scheme or photo) for connecting the FBGs to the system.
3. The authors note that an important condition for correct registration of deformations is constant temperature. I ask you to highlight in the text the measures that made it possible to do this.
4. Some sections in the article end with figures, I recommend placing them higher, after the first mention.
